# Dense Unsupervised Learning for Video Segmentation

**Nikita Araslanov**[1]  **Simone Schaub-Meyer**[1]  **Stefan Roth**[1,2]

[1]Department of Computer Science, TU Darmstadt   [2]hessian.AI
{nikita.araslanov, simone.schaub, stefan.roth}@visinf.tu-darmstadt.de

## Abstract

We present a novel approach to unsupervised learning for video object segmentation (VOS). Unlike previous work, our formulation allows to learn dense feature representations directly in a fully convolutional regime. We rely on uniform grid sampling to extract a set of anchors and train our model to disambiguate between them on both inter- and intra-video levels. However, a naive scheme to train such a model results in a degenerate solution. We propose to prevent this with a simple regularisation scheme, accommodating the equivariance property of the segmentation task to similarity transformations. Our training objective admits efficient implementation and exhibits fast training convergence. On established VOS benchmarks, our approach exceeds the segmentation accuracy of previous work despite using significantly less training data and compute power.

## 1 Introduction

Unsupervised learning of visual representations has recently made considerable and expeditious advances, in part already outperforming even supervised feature learning methods [8, 11, 14]. Most of these works, however, require substantial computational resources [8, 11], and only few accommodate one of the most ubiquitous types of visual data: videos [13, 33]. In contrast to image sets, video data embeds ample information about typical transformations occurring in nature. Exploiting such cues may allow systems to learn more task-relevant invariances, instead of relying only on hand-engineered data augmentation [30]. In this work, we propose to learn such invariances with a novel framework for the task of video object segmentation (VOS). In contrast to previous works [17, 19], we learn dense representations efficiently in a *fully convolutional* manner, previously considered prone to degenerate solutions [17]. This is achieved by leveraging the assumption of feature equivariance to similarity transformations (*e. g.*, multi-scale cropping) applied to the input frame.

Fig. 1 presents an overview of our approach. For each video sequence, we designate one of the frames as the *reference* frame. We sample a set of features, produced from the reference frame by a fully convolutional network, on a spatially uniform grid, and refer to them as *anchors*. Using a contrastive formulation [12], our approach learns to represent the temporally proximate frames to the reference in terms of these anchors. Importantly, this representation is learned to be equivariant to similarity transformations. Towards this goal, we devise a self-training objective [20] that generates pseudo labels by leveraging the second, transformed view of the original video sequence. These pseudo labels contain dominant assignments of the features from the second view to the anchors. Following the self-training mechanism, we learn to assign the features in the original view consistently with the transformed view. Our self-supervised loss further disambiguates the anchors themselves spatially and between independent video sequences, while also ensuring their transformation equivariance.

Implementing this process in a novel framework, we attain a new state of the art in video segmentation accuracy, with compelling computational and data efficiency. Our framework can be trained on

---

Code (Apache-2.0 License) available at `https://github.com/visinf/dense-ulearn-vos`.

35th Conference on Neural Information Processing Systems (NeurIPS 2021).

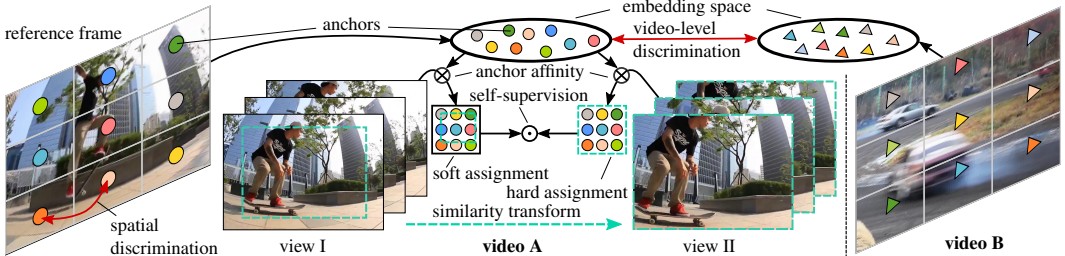

Figure 1: **Illustration of the main idea.** We learn dense visual feature representations in an unsupervised way by *(i)* discriminating the features both spatially and at the video level, and *(ii)* embedding video representations in terms of a temporally persistent set of video-specific anchors. We devise a simple and effective approach that sidesteps trivial solutions by exploiting feature equivariance to similarity transformations of the input frame, which allows to efficiently learn dense representations in a fully convolutional manner.

a single commodity GPU, yet achieves higher segmentation accuracy than previous work, while requiring orders of magnitude less data. Improving the accuracy even further with more data, our approach also exhibits favourable scalability in a more comprehensive testing scenario, which has not yet been shown in prior work [17, 38].

## 2   Related work

The research domains most relevant to our work are video object segmentation (VOS) without supervision and representation learning.

**VOS.**   The inference setting of semi-supervised VOS, which is the task considered here, is to densely track object masks provided in the first frame. It is typically approached by learning dense feature representations to establish temporal correspondences for propagating the semantic labels. While most previous methods are supervised by pixel-wise mask annotations [*e. g.*, 27], an emerging direction is to leverage unsupervised learning. Our work contributes to these research efforts. The Contrastive Random Walk (CRW) of Jabri et al. [17] and its predecessor [39] exploit a time-cycle consistency constraint, akin to the forward-backward consistency in unsupervised optical flow estimation [24]. Other methods [18, 19] learn the label propagation process at training time by substituting the target mask (which is unavailable at training time) with the image, an idea inspired by video colourisation [37]. The algorithm for label propagation itself significantly affects the VOS accuracy. Mask propagation with naive pixel-wise correspondences usually benefits greatly from leveraging more sophisticated region-level propagation [19, 21], which is not the focus here.

**Representation learning.**   Although previous work on unsupervised representation learning pursued spatially view-invariant objectives [5, 8, 11, 14, 15, 25, 36], a concurrent study [10] finds that equipping these methods with temporally-invariant, or *temporally-persistent* constraints, yields sizeable gains in terms of accuracy on action recognition benchmarks. These findings align well with previous and concurrent work tailored specifically to learning spatio-temporal representations [3, 9, 13, 32, 33] for video recognition and retrieval tasks. These works additionally distinguish the representations of frames from different videos (or, at different timesteps). The limitation of these works in the context of VOS is that they do not learn *dense*, but *global* feature representations at the image or video level, and typically require considerable computational budgets. By contrast, Pinheiro et al. [31] learn dense representations and exploit the equivariance constraint, as in our work. However, their approach is limited to image sets of static scenes, whereas we address videos of (potentially) dynamic scenes here.

**Semi-supervised representation learning.**   Recent works [13, 41] leverage optical flow [34] to ensure the representation similarity of temporally corresponding features. Such an objective can be alternatively supervised by motion segments [28]. Spatial equivariance has been also explored in part co-segmentation [16], a setup that requires semantically aligned image pairs, as well as in weakly supervised semantic segmentation [40] and unsupervised domain adaptation [1]. In contrast to this body of work, we learn to establish temporal correspondences via a suitable feature representations learned in a completely unsupervised way.

**Clustering.** Contrastive learning methods are inherently expensive due to the size of pairwise feature comparisons they consider in training [8, 14]. Clustering provides a more computationally efficient alternative [6], since pairwise distances need to be computed only *w. r. t.* a (moderately sized) set of pre-defined clusters. Combining these ideas, Caron et al. [5] propose to cluster the embeddings and then enforce consistency between the cluster assignments by contrastive learning. The main challenge in this approach is to impose an expressive prior on the cluster assignments. For example, a uniform prior prevents trivial solutions where feature assignments collapse to a single cluster.

Here, we neither explicitly define the clusters nor the prior. Instead, we sample attractor points, called *anchors*, directly from the available feature representations using a spatial grid sampling strategy.

## 3 Method

Learning feature representations from unlabelled data is an inherently ill-posed problem and requires specifying assumptions about what constitutes a useful feature embedding with a downstream task in mind. We begin with outlining the assumptions guiding our unsupervised learning method for video segmentation, before delving into the technical specifics of their implementation. We then look in more detail into the mechanisms that our method implements to comply with those assumptions.

### 3.1 Assumptions

**Assumption 1:** *Non-local feature diversity.* We assume that the distinctness of visual artefacts in a natural scene exhibits spatial correlation: the more distant two visual phenomena are on the pixel grid, the more likely they are to belong to different *semantic* entities, and vice versa. Dense representations of the images should mirror this property, *i. e.* be spatially distinguishable. Note that this assumption is *weaker* than that of spatial smoothness, which is problematic at object boundaries, where it would encourage the same representation of the object and the background. In contrast, non-local feature diversity is agnostic to the spatial arrangement of the objects and the background in the scene. On a feature grid, where every cell corresponds to its own receptive field in the image, we can discriminate the representation on all levels of the semantic hierarchy (provided sufficient grid resolution): between the objects and the background, as well as between object parts. Jabri et al. [17] implicitly relied on this assumption to spatially distinguish the nodes of the space-time graph.

**Assumption 2:** *Temporal coherence.* We posit that the feature representation extracted from one frame in a video should be closer in the embedding space (*e. g.*, in terms of the cosine similarity) to a feature vector from another temporally close frame of the same sequence, rather than to a feature embedding derived from another video. The premise for this assumption is the temporal coherence in videos: the changes in appearance within a video sequence are seldom as significant as between two independent video shots. Also leveraged by Wang et al. [38], this is a desirable property for dense tracking, which is our focus in this work.

**Assumption 3:** *Temporal persistence of semantic content.* We further assume that the semantic content of video clips remains unchanged, at least for a short time span. Specifically, if we represent a given reference frame with a set of distinct features (following Assumption 1), the semantic content of temporally close frames can be faithfully represented with the same feature set. Note that the contrastive random walk (CRW) [17] makes a somewhat stronger assumption that *every* frame in a sequence must contain the same feature set as the reference frame. By contrast, under our assumption every frame may comprise only a (proper) subset of the reference feature set. Although the edge dropout technique improves model robustness to *partial* occlusions in [17], following our assumption allows to handle *full* occlusions, provided the occluded object is visible in the reference frame.

**Assumption 4:** *Equivariance to similarity transformations.* Finally, we require the feature representation be equivariant to similarity transformations, *i. e.* scaling or flipping of an input video frame should produce a correspondingly scaled or flipped feature embedding. This assumption has not been yet explored in dense unsupervised learning from videos, and is in contrast to the common practice of unsupervised learning from static image sets, where two views produced by a random similarity transform need to yield the same global, hence *invariant* representation [8].

While Assumptions 1 and 2 have been studied independently in prior work [17, 38], we investigate their combination here, and further complement them with our occlusion-aware variant of Assumption 3 and the yet unexplored Assumption 4 on equivariance.

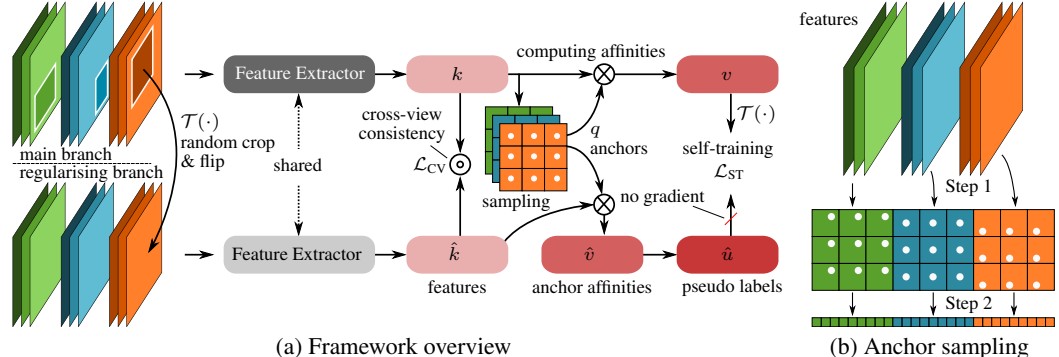

(a) Framework overview      (b) Anchor sampling

Figure 2: **Overview.** Our framework *(a)* consists of a single CNN-based feature extractor that processes video sequences in the main branch. The second regularising branch takes a transformed version (random cropping and flipping) of the videos as input and produces pseudo labels $\hat{u}$ *w. r. t.* the anchors $q$ extracted from the features of the main branch. The self-training loss $\mathcal{L}_{ST}$ learns space-time embeddings implementing our Assumptions 1–3 and, implicitly, Assumption 4 on feature equivariance. The cross-view consistency loss $\mathcal{L}_{CV}$ further facilitates the latter explicitly. To sample the anchors *(b)*, we first randomly select one frame per video sequence in the current batch (Step 1), and then sample the features at random on a uniform grid (here, with size $3 \times 3$) in Step 2 to obtain $q$.

## 3.2 Overview

The core of our training procedure is a self-supervised loss imposing four assumed properties on the feature representation. First, we learn to densely represent every video frame in terms of spatially discriminative features (Assumption 1). Second, the set of discriminative features representing a video is learned to be distinguishable from the set of another video sequence (Assumption 2). Third, our method learns to represent every frame in a video sequence as a composite of discriminative features extracted from a single reference frame of the same sequence (Assumption 3). Last, we learn a feature representation satisfying these properties under the equivariance constraint (Assumption 4). Specifically, we minimise the distance between every pair of spatially corresponding features extracted from two views and related by a similarity transform *w. r. t.* non-corresponding pairs.

## 3.3 Framework

The feature extractor in our framework, illustrated in Fig. 2a, is a fully convolutional network [22]. It processes two copies of the input image batch, comprising the same set of video sequences, in the *main* and *regularising* branches. The purpose of the regularising branch is to prevent degenerate solutions observed in previous attempts to learn feature representations in a fully convolutional manner [17]. While the main branch receives the original video frames, we feed a transformed version of these frames to the regularising branch. Specifically, we extract random multi-scale crops and apply horizontal flipping at random. We denote this random similarity transform as $\mathcal{T}(\cdot)$. Both the main and the regularising branch produce dense $L_2$-normalised $K$-dimensional feature tensors $k, \hat{k} \in \mathbb{R}^{B,T,K,h,w}$, respectively, where $\|k_{i,j,:,l,m}\| = \|\hat{k}_{i,j,:,l,m}\| = 1$, $B$ is the batch size, $T$ is the number of selected frames from a video sequence, and $h, w$ are the spatial dimensions of these embeddings. Terms related to the regularising branch are denoted with the hat notation ( ˆ ) in the following. In the next step, we compute pairwise feature distances to leverage contrastive learning [12]. Recall from Sec. 3.2 that our goal is to discriminate features *(i)* spatially within individual frames under the equivariance constraint, and *(ii)* temporally, to represent each frame in a video sequence in terms of the same feature set while distinguishing between independent video sequences.

**Anchor sampling.** We take a bi-level sampling approach based on clustering for improved training efficiency, illustrated in Fig. 2b. First, for each video sequence in the batch, we sample one frame at random. We define it as the *reference* frame for the video sequence from which it originates. Relying on Assumption 3 that the frames of the same sequence share semantic content, we leverage the randomly chosen reference frame for extracting a set of *video-level* features, *i. e.* encouraged to be shared between temporally close video frames. In more detail, instead of computing the pairwise distances between every feature vector in the batch, we define a spatially uniform grid of size $N \times N$

on the feature tensor of the reference frame, and draw one sample per grid cell. This is to collect features that are spatially distinct (Assumption 1) and cover the full image. As a result, we obtain $(B \times N^2)$ $K$-dimensional feature embeddings, which we will refer to as *anchors* and denote as $q$. Defining this grid sampling operation as $\mathcal{G}_N(\cdot)$, we can write this step as $q := \mathcal{G}_N(k)$. We then compute pairwise distances between the features $k$ and $\hat{k}$ w.r.t. these anchors, rather than to the features themselves. The outcome is a distance matrix of size $(B \cdot T \cdot h \cdot w) \times (B \cdot N^2)$. Compared to dense sampling, this reduces memory and the computational budget by a factor of $O(T \cdot h \cdot w \,/\, N^2)$.

**Computing affinity to anchors.** Following our cluster-based analogy, we compute affinities of the features $k$ and $\hat{k}$ from the two branches w.r.t. the anchors $q$. Note that we extract the anchors only from the main branch and share them with the regularising branch. Since the anchors $q \in \mathbb{R}^{BN^2 \times K}$ are sampled from $k$, they also have unit $L_2$-norm, i.e. $\|q_{i,:}\| = 1$ for all $i$. Let $v, \hat{v} \in \mathbb{R}^{BThw \times BN^2}$ denote the affinities of $k$ and $\hat{k}$ w.r.t. $q$, respectively, which we compute using a softmax over cosine similarities w.r.t. all anchors, i.e.

$$v_{i,j} = \frac{\exp(k_i \cdot q_j / \tau)}{\sum_l \exp(k_i \cdot q_l / \tau)}, \qquad \hat{v}_{i,j} = \frac{\exp(\hat{k}_i \cdot q_j / \tau)}{\sum_l \exp(\hat{k}_i \cdot q_l / \tau)}, \tag{1}$$

where $\tau \in \mathbb{R}^+$ is a scalar temperature hyperparameter. Put simply, $v_{i,j} \in [0,1]^{BThw \times BN^2}$ represents the similarity between feature $k_i$ and anchor $q_j$ (analogously for $\hat{v}_{i,j}$).

**Space-time self-training.** We first generate pseudo labels of anchor assignments for self-training by acquiring the dominant anchor index for each feature from the regularising branch,

$$\hat{u}_i = \arg\max_{j \in \mathcal{N}(i)} \hat{v}_{i,j}, \tag{2}$$

where $\mathcal{N}(i)$ is the index set of the anchors that stem from the same video clip as the feature vector with index $i$. Observe that $v$ and $\hat{u}$ are in spatial correspondence via a similarity transformation $\mathcal{T}$. The former represents *soft* assignments of the feature vectors from the original view to the anchors, while the latter contains *hard* assignments of the features from the regularising branch. Our primary self-supervised loss minimises the distance of the features extracted from the other temporally close frames to the anchors,

$$\mathcal{L}_{ST} = -\sum_{i \notin \mathcal{R}} \log \mathcal{T}(v_{i,\hat{u}_i}), \tag{3}$$

where $\mathcal{R}$ is the index set of the features extracted from the reference frames; $\mathcal{T}(\cdot)$ spatially aligns the soft affinities $v$ of the main branch with the pseudo labels obtained with the help of the regularising branch. As we analyse in Sec. 3.4, this loss term relates to the spatial and temporal properties of the learned representation *within the same view* (since we do not propagate the gradient in Eq. (2) following our self-training approach), hence we refer to this objective as *space-time self-training*. However, implementing Assumptions 1–3, this objective only implicitly fulfils Assumption 4 on equivariance by means of seeking a consistent anchor assignment between the two views.

**Cross-view consistency.** To generate the pseudo labels in Eq. (2), we assume that the dominant assignment of $\hat{k}$ to the anchors $q$ is meaningful, despite the two representations originating from different synthetic views (related by $\mathcal{T}(\cdot)$). By following our space-time self-training in Eq. (3), we already find this assumption largely to hold for datasets of modest size (*cf.* Sec. 4.3). However, we observed additional accuracy benefits and improved training stability from *explicitly* facilitating the equivariance, our Assumption 4, especially when training on larger datasets. We impose a cross-view consistency term on the reference features only. Note that it complements the self-supervision with the pseudo labels in Eq. (3), which omits the reference features. Re-using our grid sampling operator $\mathcal{G}_M(\cdot)$ parametrised by a cross-view grid size $M$, we subsample the features from the main and the regularising branches as $r = \mathcal{G}_M(\mathcal{T}(k))$ and $\hat{r} = \mathcal{G}_M(\hat{k})$ such that $r, \hat{r} \in \mathbb{R}^{BM^2 \times K}$. Similarly to Eq. (1), we compute pairwise affinities between these features:

$$h_i = \frac{\exp(r_i \cdot \hat{r}_i / \tau)}{\sum_{l \neq i} \exp(r_i \cdot \hat{r}_l / \tau)}. \tag{4}$$

This affinity contrasts the cosine similarity between the corresponding features (since $r$ and $\hat{r}$ are spatially aligned) w.r.t. non-corresponding pairs. We define the cross-view loss term to uphold this correspondence in the embedding space by minimising

$$\mathcal{L}_{CV} = -\sum_{i \in \mathcal{R}} \log h_i. \tag{5}$$

A hyperparameter $\lambda$ weights its contribution to the total loss used for training our framework:

$$\mathcal{L} = \mathcal{L}_{ST} + \lambda \mathcal{L}_{CV}. \tag{6}$$

### 3.4 Analysis

We now take detailed look at how our framework implements the assumptions from Sec. 3.1.

**Non-local feature diversity (Assumption 1).** Our loss disambiguates feature representations by clustering, with the anchors acting as attractors. Recall that our formulation of the feature distance to the anchors in Eq. (1) is contrastive, *i. e.* it is measured relative to all other anchors. As a result, minimising such distance would not only encourage an increased cosine similarity of the features to the anchors, but also a *decreased* cosine similarity between the anchors themselves. Recall that the anchors are a subset of features sampled spatially from a randomly selected video frame on a uniform grid. Consequently, our loss learns to discriminate between spatially distinct features.

**Temporal coherence (Assumption 2).** In Eq. (1), we compute the affinity of the features to the anchors extracted from multiple videos in the training batch. Note that we select the dominant anchors in Eq. (2) for self-training only from the same video sequence as the feature itself. This implies that *(i)* the features will be attracted only to the anchors originating from the same video, and *(ii)* the distance between the anchors and the features from different video sequences will increase by virtue of our contrastive formulation of the affinity. Note that the latter also implements inter-video discrimination, analogous to instance-specific learning from image sets, which was shown to result in class-specific representations [*e. g.*, 8].

**Temporal persistence of semantic content (Assumption 3).** Our anchor sampling strategy ensures that all anchors stem from a single video frame, the reference frame. By design of the pseudo labels (*cf.* Eq. (2)), our self-training loss in Eq. (3) aligns the representation of the other frames only with the reference originating from the same video.

**Equivariance (Assumption 4).** We generate a random similarity transformation to synthesise the input to the regularising branch. The task of video object segmentation (VOS), studied here, is equivariant under this family of transformations: flipping or scaling the image should result in a corresponding change in the segmentation output. Our cross-view consistency loss in Eq. (5) explicitly facilitates this property.

### 3.5 Discussion

**Preventing degenerate solutions.** In our preliminary experiments without the regularising branch and Assumption 4, we observed that the model would rapidly converge to a trivial solution, previously also reported by Jabri et al. [17]. Our investigation suggested that the network was encoding positional cues into a degenerate feature representation. We hypothesise that such solutions may be a consequence of the limited spatial invariance in contemporary CNN implementations [2] and the widely adopted padding. The padding type is, in fact, irrelevant, since any deterministic padding strategy provides a stable and predictable pattern. We also found it necessary to spatially jitter the grid for sampling the anchors, rather than to maintain a fixed offset, to prevent further shortcut solutions.

**Computational scalability.** Using a grid sampling strategy to extract anchors allows for considerable computational savings. For example, if $h \times w = 32 \times 32$, and we use a grid of size $N \times N = 8 \times 8$, this will have a computational and memory saving factor of 16. This saving becomes even more valuable if we scale up the storage costs of the affinities $v$ and $\hat{v}$, which have the size of $B \cdot T \cdot h \cdot w \times B \cdot N^2$, *w. r. t.* $B$ and $T$. Improving the computational footprint with such subsampling is non-trivial in MAST [19] due to the use of the dense reconstruction loss, and in CRW [17] as it extracts image patches with 50% overlap, hence propagates the same pixels through the network up to three times.

**Practical considerations.** The order of the frames from a sequence in the batch is irrelevant and can be randomly selected, since we do not require any assumptions about motion continuity. Further, we do not use any momentum network or queue buffers, *e. g.* such as in [14]. Our training is surprisingly stable without any of them. Further, we only use similarity transformations to augment the training data, but no appearance-based augmentations such as photometric noise [8, 11, 14]. Instead, we rely on learning natural changes in the appearance directly from video data.

**Comparison to Caron et al. [5].** In contrast to [5], the representation of our clusters (anchors) is *non-parametric*. Specifically, we do not learn a global set of cluster features shared by the complete video dataset. The advantage is that we neither need to set the number of clusters *a priori*, nor to specify a prior on the cluster assignment.

Table 1: **Video object segmentation quality** on DAVIS-2017 validation in terms of the mask ($\mathcal{J}$) and boundary ($\mathcal{F}$) accuracy (IoU). The subscript $[\cdot]_r$ denotes the recall of the metric, while $[\cdot]_m$ signifies the mean. $\times 2$ denotes an output stride of 4 instead of 8, *i. e.* the spatial feature resolution is twice as large. $\# / t$ details dataset characteristics: the number of videos / the total dataset duration in hours.

| Method | $\times 2$ | Train Data | $\# / t$ | $\mathcal{J}_m$ | $\mathcal{J}_r$ | $\mathcal{F}_m$ | $\mathcal{F}_r$ | $\mathcal{J}\&\mathcal{F}_m$ |
|---|---|---|---|---|---|---|---|---|
| Baseline (random initialisation) | — | — | — | — | — | — | — | 43.1 |
| CorrFlow [18] | — | | | 48.4 | 53.2 | 52.2 | 56.0 | 50.3 |
| MAST [19] | ✓ | OxUvA | 366 / 14 | 61.2 | 73.2 | 66.3 | 78.3 | 63.7 |
| Ours | — | | | **63.4** | **76.1** | **67.2** | **79.7** | **65.3** |
| MAST [19] | ✓ | YT-VOS | 4.5K / 5 | 63.3 | 73.2 | 67.6 | 77.7 | 65.5 |
| Ours | — | | | **67.1** | **81.2** | **71.6** | **84.9** | **69.3** |
| ContrastCorr [38] | — | TrackingNet | 30K / 140 | 60.5 | — | 65.5 | — | 63.0 |
| Ours | — | | | **67.1** | 80.9 | **71.7** | 84.8 | **69.4** |
| CRW [17] | — | Kinetics-400 | 300K / 833 | 64.8 | 76.1 | 70.2 | 82.1 | 67.6 |
| Ours | — | | | **66.7** | **81.4** | **70.7** | **84.1** | **68.7** |

**Comparison to CRW [17].** The implementation of our approach is simpler, since the frame ordering within the batch is irrelevant, and more computationally efficient, as we compute feature affinities in parallel rather than sequentially (see Table 3 for detail). We also use a weaker assumption on semantic persistence (Assumption 3), which can be advantageous in video sequences with occlusions. Additionally, we learn to discriminate features between different videos, which is not part of CRW.

# 4 Experiments

To evaluate the learned feature representations, we conduct experiments in the setting of semi-supervised VOS. The task provides a set of segmentation masks for the first frame in a video sequence and requires the evaluated algorithm to densely track the demarcated objects in the remaining frames. We largely follow the VOS setup of Jabri et al. [17] and evaluate our method on DAVIS-2017 [35]. Following Lai et al. [19], we additionally test our approach on the YouTube-VOS *val* by submitting our results to an evaluation server [42].

**Implementation details.** Similarly to Jabri et al. [17], we use ResNet-18 as the backbone network for our feature extractor. We evaluate the correspondences using the output of the fourth (the last) residual block. To obtain the output stride of 8 in our feature extractor, we remove the strides of the last two residual blocks, as in previous work [17, 38]. At training time, we first scale the video frames such that the lowest side is between 256 and 320 pixels, and extract random crops of size $256 \times 256$. We train our network with Adam and the learning rate $10^{-4}$ on the smaller YouTube-VOS and OxUvA [35], whereas we found SGD with the learning rate $10^{-3}$ to work better on the larger Kinetics-400 and TrackingNet datasets. We set the temperature $\tau = 0.05$ throughout our experiments; we observed its influence on the accuracy to not be significant. The hyperparameter $\lambda$, trading off the influence of the cross-view consistency, equals $0.1$ by default, and we empirically study its role in Sec. 4.3. We train our models on one $A100$ GPU, although training our most accurate configuration of the framework requires only 12GB of memory, hence a *single* Titan X GPU is actually sufficient.

**Label propagation.** To propagate the semantic labels from the initial ground-truth annotation, we rely on the label propagation algorithm implemented by Jabri et al. [17], detailed in Appendix C. In particular, for every feature embedding in the current frame, we compute its cosine similarity *w. r. t.* the features in the previous $N_T = 20$ context frames and the first reference frame. We select $N_K = 10$ feature embeddings with the highest similarity, divide the similarity by $\tau$ and compute a softmax to obtain normalised affinity values. The mask prediction for this feature embedding is a convex combination of the mask predictions of its neighbours, weighted by this affinity.

## 4.1 Evaluation on DAVIS-2017

To evaluate on DAVIS-2017 [29] *val*, we independently train our feature extractor on 4 datasets. The OxUvA dataset [35] spans 366 video sequences with a total duration of 14 hours. The second

dataset is YouTube-VOS [42], which by comparison to OxUvA contains more videos (around 4.5K sequences), but has a shorter overall duration of 5.6 hours. In addition, we train on larger datasets, TrackingNet [26] and Kinetics-400 [7]. These datasets contain significantly more video sequences, albeit of lower resolution and quality (*e. g.*, due to compression artefacts). While running the inference on DAVIS-2017 [29], we process the frames at a resolution of $480p$ for fair comparison.

Table 1 reports the segmentation accuracy in terms of two types of metrics: The Jaccard's index measures the intersection-over-union (IoU) of the object mask respectively the contour; we report the mean of mask and contour IoU, $\mathcal{J}_m$ and $\mathcal{F}_m$, as well as the recall $\mathcal{J}_r$ and $\mathcal{F}_r$ (with an IoU threshold of 0.5) [29]; $\mathcal{J}\&\mathcal{F}_m$ is the average of $\mathcal{J}_m$ and $\mathcal{F}_m$. In a comparable setting where all methods use the same dataset for training, our approach clearly improves over the state-of-the-art accuracy. When trained on OxUvA, our approach outperforms MAST [19] by $1.6\%$ $\mathcal{J}\&\mathcal{F}_m$. This improvement is even more pronounced when trained on YouTube-VOS, where we outperform MAST [19] by $3.8\%$ in $\mathcal{J}\&\mathcal{F}_m$ score. This improvement is especially notable, since MAST [19] produces its feature embeddings at twice the resolution of our method, hence has a larger memory footprint. These results are consistent when training our approach on larger but less carefully curated datasets. Using TrackingNet, we improve over the previous result of Wang et al. [38] by $6.4\%$ $\mathcal{J}\&\mathcal{F}_m$. Compared to CRW [17], our approach reaches a higher $\mathcal{J}\&\mathcal{F}_m$ score by $1.1\%$. Remarkably, using the smaller YouTube-VOS dataset for training, our method already achieves comparable or higher VOS accuracy compared to previous work [17, 38] that required significantly larger datasets for training. We also observe that the diversity and the quality of the dataset, *i. e.* the number of the videos and their resolution, tends to improve the quality of our learned features in terms of the segmentation accuracy.

## 4.2 Evaluation on YouTube-VOS

Following Lai et al. [19], we additionally evaluate our features on the YouTube-VOS 2018 *valid* split. With 474 video sequences containing 91 unique object classes, its diversity represents a significantly more challenging and comprehensive test bed for VOS than DAVIS-2017. We train our model on three datasets as before: YouTube-VOS, TrackingNet, and Kinetics-400. The *train* set of YouTube-VOS contains only some of the object classes present in *valid* and the benchmark distinguishes between these as "seen" and "unseen" categories. We submit our results to the official evaluation server to obtain the quantitative metrics $\mathcal{J}_m$ and $\mathcal{F}_m$. YouTube-VOS *valid* has additional specifics compared to the DAVIS benchmark and previous work tends to use a different label propagation strategy to address them (*e. g.*, the initial object masks may appear in different intermediate frames, rather than only in the first frame at once). Since we use the label propagation from CRW [17], we also evaluate this model trained on Kinetics-400 for a fair comparison. Note that the label propagation of MAST [19] benefits from a larger feature resolution and a more advanced two-stage inference process: first detecting a ROI and then computing feature correlations bounded by the ROI.

The results in Table 2 show that our approach sets a new state of the art, improving over CRW [17] by $0.8\%$ mean score. In a comparable setting when our method and CRW use the same training data, Kinetics-400, we improve by $0.7\%$. With larger training datasets, Kinetics and TrackingNet, we reach higher VOS accuracy compared to training on the smaller YouTube-VOS. Such scalability has not been shown in previous work [17, 38], where the training dataset was limited to a single instance, or two instances of comparable (small) scale [19]. Notably, the accuracy from training on YouTube-VOS already matches that of CRW when trained on much more data. Finally, we also outperform fully supervised methods, OSVOS [4] and PreMVOS [23], especially on unseen categories.

## 4.3 Further analysis

We study *(i)* the interplay between the number of video sequences $B$ per batch and the number of frames $T$ per video; *(ii)* the influence of the cross-view loss term via the trade-off hyperparameter $\lambda$; *(iii)* the framework's sensitivity to the size of the sampling grid of the anchors $N$ and *(iv)* cross-view features $M$; *(v)* increasing the temporal window of the video clips in training. We conduct all experiments by training the corresponding model on YouTube-VOS. Fig. 3 visualises the results in terms of the $\mathcal{J}\&\mathcal{F}_m$ score on DAVIS-2017 (val). Our baseline configuration, depicted as the centre node, corresponds to the default setting of $B = 16, T = 5, \lambda = 0.1, N = 8, M = 4$. Every edge of the same style and colour represents a change of selected hyperparameters, while keeping the default values for the rest. To put the significance of the differences into perspective, we also compute the standard deviation of the $\mathcal{J}\&\mathcal{F}_m$ score on DAVIS-2017 across 5 training runs, amounting to $\pm 0.42$.

Table 2: **Results on YouTube-VOS 2018 (val).** We compare our models pre-trained on YouTube-VOS, TrackingNet, and Kinetics to previous work. CRW [17] and Colorize [37] were trained on Kinetics-400; CorrFlow [18] and MAST [19] used OxUvA and YouTube-VOS, respectively. CRW [17] uses the same label propagation as ours.

| Method | Seen | | Unseen | | Mean |
|---|---|---|---|---|---|
| | $\mathcal{J}_m$ | $\mathcal{F}_m$ | $\mathcal{J}_m$ | $\mathcal{F}_m$ | |
| *Unsupervised* | | | | | |
| Colorize [37] | 43.1 | 38.6 | 36.6 | 37.4 | 38.9 |
| CorrFlow [18] | 50.6 | 46.6 | 43.8 | 45.6 | 46.6 |
| MAST [19] | 63.9 | 64.9 | 60.3 | 67.7 | 64.2 |
| CRW [17] | 68.7 | 70.2 | 65.4 | **75.2** | 69.9 |
| Ours-YouTubeVOS | 69.6 | 71.3 | 65.0 | 73.5 | 69.9 |
| Ours-Kinetics | 69.9 | 71.3 | **66.5** | 74.8 | 70.6 |
| Ours-TrackingNet | **70.2** | **71.9** | 66.3 | 74.5 | **70.7** |
| *Supervised* | | | | | |
| OSVOS [4] | 59.8 | 60.5 | 54.2 | 60.7 | 58.8 |
| PreMVOS [23] | 71.4 | 75.9 | 56.5 | 63.7 | 66.9 |
| STM [27] | **79.7** | **84.2** | **72.8** | **80.9** | **79.4** |

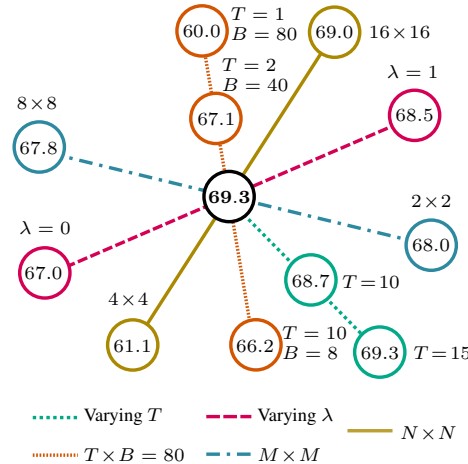

Figure 3: **Ablation study.** We report $\mathcal{J}\&\mathcal{F}_m$ on DAVIS-2017 (val) by varying the hyperparameters of our baseline configuration (centre): $B = 16$ (no. of videos in a batch), $T = 5$ (no. frames per video), $\lambda = 0.1$ (cross-view trade-off), $N = 8$ (anchor grid size), $M = 4$ (cross-view grid size). See Sec. 4.3 for details.

**Balance between the number of videos and their size.** The effective batch size of our method is $B \times T$, hence depends on the number of videos and their duration. Here, we investigate the balance between the size of the video set $T$ and the number of distinct video sequences $B$ used in a single training batch. We fix $B \times T$ to $80$ as the invariant, and evaluate three additional configurations of $B$ and $T$. Our main observation in Fig. 3 is that *using multiple frames,* i. e. *ensuring $T > 1$, is crucial*: training with only a single frame per video ($T=1$) reduces the quality of learned feature representations significantly, by $9.3\%$ $\mathcal{J}\&\mathcal{F}_m$. In light of Assumption 3, this is expected, as only in a multi-frame setting our model can learn a shared representation of the video set.

**The influence of the cross-view consistency $\gamma$.** By setting $\gamma = 0$, we train our approach without the cross-view consistency (*cf.* Eq. (5)). We find that this results in an accuracy drop of $2.3\%$ $\mathcal{J}\&\mathcal{F}_m$, confirming its accuracy benefits. On the other extreme, we increase its values from $0.1$ to $1.0$ and find it dominating over the space-time self-training term, hence inhibiting learning of temporal coherence: $\mathcal{J}\&\mathcal{F}_m$ decreases by $0.8\%$. Overall, we confirm that the benefits of data augmentation used via cross-view consistency is only complementary to our self-training. This is in contrast to the pivotal role of data augmentation in prior work [*e. g.*, 5, 8, 11]. Our main hypothesis, elaborated in Appendix B.2, is that artificial image transformations do not reflect the transformations occurring in time in video sequences, which is important to establish accurate *temporal* correspondences.

**Grid resolution.** We vary the size of the sampling grid for the anchors $N$ and the cross-view consistency $M$ (Eq. (5)). Since a high grid density (*i. e.* large $N$ and $M$) implies an increased computational footprint (*cf.* Sec. 3.5), we seek to reduce it without compromising VOS accuracy. A low grid resolution may be too coarse in terms of discriminating a sufficient level of detail. Conversely, a high grid resolution may focus the learning more on discriminating between object parts, which may be detrimental to object-level disambiguation. We find that both of these extremes lead to a tangibly lower VOS accuracy, *e. g.* using $N = 4$ leads to a drop in $\mathcal{J}\&\mathcal{F}_m$ score by $8.2\%$. Varying $M$ leads to a more moderate decrease in $\mathcal{J}\&\mathcal{F}_m$, by at most $1.5\%$ for $M = 8$. Regarding the discrepancy between the optimal $N = 8$ and $M = 4$, we hypothesise that while a grid size of $N = 4$ may be sufficient to satisfy our Assumption 1, it may undermine Assumption 3, since a sparser grid may miss feature vectors representing the object of interest, visible in the temporally close frames.

**Sampling more frames $T$ per video.** We now fix $B = 16$ and increase the number of frames per video $T$. Higher $T$ have the appeal of providing larger degrees of natural transformations. However, larger videos also contain more occlusion, which our method can handle, but also disocclusions, which violate our Assumption 3 on temporal persistence, hence currently problematic. The results

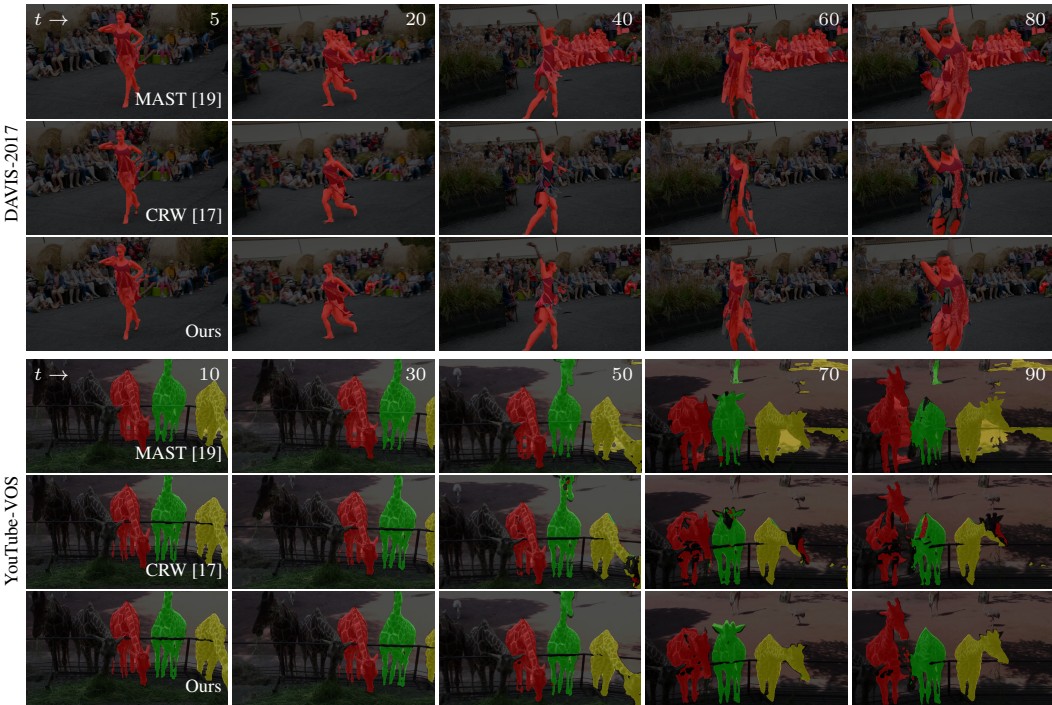

Figure 4: **Qualitative examples on DAVIS-2017 and YouTube-VOS.** The representation learned by our method is robust to occlusions, naturally occurring non-rigid transformations and deals well with object-background disambiguation.

confirm this expectation: using more frames does not further improve the VOS accuracy. For example, increasing $T$ to 15, we match $\mathcal{J}\&\mathcal{F}_m$ of the baseline, yet at additional computational costs.

**Qualitative examples.** Fig. 4 shows examples of mask propagation, leveraging our learned feature representation. We observe temporal correspondences to remain stable and accurate despite complex self-occlusions (*e. g.*, torso of dancer), non-rigid deformations, and appearance changes due to shadows (*e. g.*, giraffes). Our method does not exhibit "bleeding" artefacts of MAST [19] and produces more complete segmentation masks than [17] despite using less data for training (YouTube-VOS for the DAVIS-2017 example, TrackingNet for the YouTube-VOS result). See supplemental for more visual results.

Table 3: **Computational efficiency.**

| Method | # | $\Delta t$ | Mem |
|---|---|---|---|
| ContrastCorr [38] | 0.4M | 1d | 44GB |
| Ours-TrackingNet | **0.2M** | 1d | **12GB** |
| CRW [17] | 2M | 7d | 22GB |
| Ours-Kinetics-400 | **0.3M** | **2d** | **12GB** |
| MAST [19] | 2M | — | 22GB |
| Ours-YouTube-VOS | **0.1M** | **6h** | **12GB** |

**Computational efficiency.** Table 3 summarises the computational benefits of our approach *w. r. t.* previous work. Regardless of the training data, our approach requires less time for convergence, both in terms of the number of training iterations (#) and wall clock duration $\Delta t$. This improvement comes with modest memory requirements: to train our framework we require only 12GB of GPU memory, a twofold decrease compared to prior work.

## 5    Conclusion

We presented a simple and computationally efficient unsupervised method for learning dense space-time representations from unlabelled videos. Our approach exhibits fast training convergence and compelling data efficiency. We achieve VOS accuracy surpassing previous results despite using only a fractional amount of the training data required in prior work. We recognise the possibility of our research results to be used with malicious intents, such as for unauthorised surveillance. However, as a product of fundamental research with low technology readiness levels of 1 to 3, this is highly unlikely. Our method is yet unable to handle disocclusions. We are excited to explore such capability to improve learning a wider spectrum of invariances by leveraging larger temporal windows in videos containing complex (ego-)motion, where disocclusions are more likely to occur.

## Acknowledgments and disclosure of funding

This project has received funding from the European Research Council (ERC) under the European Union's Horizon2020 research and innovation programme (grant agreement No. 866008). This work has also been co-funded by the LOEWE initiative (Hesse, Germany) within the emergenCITY center.

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
