# OpenReview forum: "Dense Unsupervised Learning for Video Segmentation"
_NeurIPS.cc/2021/Conference — NeurIPS 2021 Poster_

### Official Review · Reviewer_ckf7 · 2021-07-05

**Rating:** 3
**Confidence:** 5

**Summary:**

This paper aims at unsupervised feature learning from videos. The learned representation is tested in a video segmentation/tracking setting on the DAVIS and the Youtube-VOS datasets. Although the paper describes many principles for the learning strategy, I could not find any of them to be new and also the results do not indicate that something fundamental was added.

**Ethical Concerns:**

No concerns

**Limitations And Societal Impact:**

The conclusions include some superficial statements, which is sufficient for this type of paper.

**Main Review:**

In 3.1. the paper introduces 4 assumptions to derive constraints for feature learning. However, all of them are well-known. The first one is spatial smoothness (which is, btw, wrong at object boundaries). The second and third assumptions look identical to me and introduce temporal smoothness. Also this has been commonly used for contrastive video representation learning. The last one is equivariance to scaling and flipping implemented by an augmentation procedure. Also this is pretty standard. Another highlighted concept is subsampling (motivated by spatial and temporal smoothness). I cannot see anything spectacularly new here either.

In 3.6. there is a discussion stating some questionable advantages of the proposed approach: Why is it good to ignore temporal causality? Common sense tells us that features generalize better if they model a causal process. Why is it good if clusters are instance-specific and do not span across instances? Establishing links across instances should improve generalization (though admittedly this is not relevant for video tracking). Scalability is bound to subsampling. What prevents other methods from subsampling as well to improve scalability? There is a price to pay, of course: as soon as the grid gets to coarse, there is relevant information loss (see ablation in Table 2b).

The efficiency argument is not sufficiently quantified by experiments.

In conclusion, I cannot say what this paper contributes. It is a relatively complicated description of well-known facts and procedures. Thus, I cannot recommend accepting the paper.

**Time Spent Reviewing:**

3h

---

> ### Author Response · Authors · 2021-08-10
> **Response to Reviewer ckf7**
>
> Thank you for your valuable and thoughtful feedback.
>
> We would like to clarify potential misunderstandings about our assumptions, which we will make clearer in the revision:
>
> * Our Assumption 1 (ll. 72-76) is the converse of spatial smoothness.
> Rather than encouraging feature representations of neighbouring pixels to be the same, Assumption 1 dictates that spatially distant pixels should be distinguishable in the feature space.
> It is precisely at the object boundaries where our assumption is more advantageous.
> Spatial smoothness would encourage the same representation of the object and the background (correctly pointed out as incorrect).
> However, following our assumption would ensure that the features of sufficiently distant pixels are distinguishable: if two features are on the opposite sides of the object boundary, we will directly discriminate the object from the background; otherwise, we merely discriminate the pixels located within the object (or the background).
>
> * Assumptions 2 and 3 (ll. 77-88) are fundamentally different.
> While Assumption 2 specifies that a video clip should be representable in terms of a small set of features (referred to as "anchors", or "prototypes"), Assumption 3 posits that these sets of features should be distinguishable between any pair of independent video clips (i.e. different sequences in the dataset).
>
> We also do not claim novelty of our assumptions. In particular, we agree that Assumption 4 on model equivariance is standard in the literature on contrastive learning.
> However, our implementation of this assumption in the framework is different at core.
> Instead of explicitly minimising the distance of feature representations between two views, as common in the literature, we learn equivariant grouping of features.
> That is, if a set of features form a cluster in one view, they should form a cluster in another view. Different from [3], however, our framework allows the two clusters to remain distinct, since it does not propagate the gradient to the regularising branch (c.f. Fig. 2), but minimises the feature distance to the anchors extracted from the same view of the main branch as the features themselves.
> This is a more data-driven approach: only the features from the same view may be directly attracted (following Assumption 2), rather than from two views generated by heuristically specified augmentations (e.g. scale) which may not model natural (or useful) transformations.
> Moreover, our empirical results in Tab. 1-3 clearly show that this approach also prevents shortcut solutions that impede learning dense representations with fully convolutional strategies (see Appendix C in [13]).
>
> > Why is it good to ignore temporal causality?
>
> This is not our claim. In fact, we agree that temporal causality is an important cue.
> In ll. 217-226 we merely point out that implementation of our approach is simpler, since the ordering of the frames within the batch is irrelevant.
> Please note that even CRW [13] is agnostic to the direction of time: the set of affinity matrices computed between the frames would be the same if the frame order is reversed.
> We will revise the text to make this point clearer.
>
>
> > Why is it good if clusters are instance-specific and do not span across instances? [ll. 215-216]
>
> This is in line with our target use case in this work: video object segmentation. Note that instance-specific discrimination is not uncommon in contrastive learning from image sets and leads to emergence of class-specific representations [5,6,9,11,35].
>
> > What prevents other methods from subsampling as well to improve scalability?
>
> Our discussion on scalability in ll. 200-204 pertains to memory efficiency.
> Improving memory efficiency of previous methods [13,15] by subsampling (or other means) is non-trivial.
> MAST [15] employs reconstruction loss and even requires a lower output stride (4), thereby increasing the memory footprint of the baseline network by roughly four times.
> Contrastive Random Walk (CRW) [13] defines a grid (7x7) of patches to represent nodes in the graph.
> Although it may appear similar to our subsampling, the main motivation behind it is completely different: to prevent shortcut solutions by restricting the receptive field in a controlled fashion.
> Since the patches are sampled with 50% overlap, it incurs additional computational overhead $-$ the same pixels need to be propagated through the network up to four times.
> Our framework obviates shortcut solutions by design without these extra costs, and employs subsampling to additionally improve memory efficiency.
>
>
> > Efficiency not sufficiently quantified.
>
> We discuss memory savings owing to grid sampling (ll. 116-121, 132-134, 200-204) and the evaluation-only mode of the regularising branch (ll. 159-162).
> We also report the training dataset specifics in Tab. 1, and compare the convergence speed and hardware requirements to previous work in the supplementary material (ll. 15-27); we will make it more prominent in the main text.
> Nevertheless, we welcome any specific suggestions for additionally quantifying the efficiency of our framework.
>
>
> **Contribution summary**
>
> We agree that portions and variants of our assumptions from Sec. 3.1 have been used in previous work, some only implicitly, and we do not claim their novelty.
> However, our proposed implementation of this particular set of assumptions is novel to our knowledge, and carefully designed for data efficiency, model simplicity and low computational footprint.
> We note that research efforts to reduce computational complexity of contrastive learning have proved particularly impactful in the past [6,9,11].
> Our work contributes to these efforts by learning dense feature representations with contrastive learning in a fully convolutional regime overcoming degenerate solutions and with less data, which has not been possible before.

---

### Official Review · Reviewer_cHxN · 2021-07-16

**Rating:** 6
**Confidence:** 4

**Summary:**

This paper proposes a method for self-supervised pretraining on video:

Given a batch of video clips, dense per-frame features are extracted from a backbone architecture like ResNet. The objective is to apply a standard contrastive penalty where a softmax loss picks out a positive pair of feature embeddings amidst a large set of negatives. Rather than compare every feature vector to every other vector in the batch (across samples, time, and space) a subset of representative vectors are chosen for comparison. Starting from BxTxHxW features, we choose from each sample a single frame and a subsampled grid of values to get BxNxN "anchors" to compare to. The full set of BxTxHxW features will each be compared to all of the sampled BxNxN anchors.

To apply the loss, for each feature we need one of the comparison anchors to serve as a positive target. This is done by finding the closest match to features from an augmented version of the same clip. That is, another inference pass of the backbone takes in a randomly resized/cropped/flipped version of the clip. Those features are compared to the anchors to find the argmax closest match. This creates a pseudo-ground truth label used to supervise the original features. Importantly, the gradient is stopped so nothing is backpropped through the pseudo-label generation.

Pretraining is done with the proposed set up and evaluated on the downstream setting of unsupervised mask propagation on DAVIS and YT-VOS.

**Limitations And Societal Impact:**

Promising to not provide advice for anyone attempting to use the proposed research in a malicious way is not exactly the strongest point, but overall the authors are correct that this is pretty basic early-stage research with few specific, harmful downstream applications.

**Main Review:**

The good:

- I think the idea itself is interesting and sufficiently innovative over the large body of emerging contrastive learning work. The idea of using the regularizing branch to choose pseudolabels makes sense to me, we've seen other work that shows a simple strategy of stopping gradients from the augmented branch will get you far [6]. Thanks to using video + dense spatial features, all the comparisons in the contrastive loss are between features from a single branch which is different from most existing contrastive methods though offers some similarity to [3] (acknowledged by the authors).
- Overall the implementation and replication of this idea seems like it wouldn't be too much trouble, there's enough detail in the paper to pull it all together, and there's a degree to which it is simpler compared to prior video methods that rely more directly on correspondence/flow.
- The ablations inform how to approach some of the critical design decisions of the method (how many frames should be used, how much to subsample the grid)

Concerns:

- It took me quite a number of read throughs to finally understand what the method was. While it may be a reflection of my own reading comprehension, I think that the paper would benefit from a clearer presentation. Perhaps some reworking of Figure 2 might help.
- The most direct comparison is to [13] which offers similar motivation + set up + downstream evaluation, but the proposed method underperforms on both DAVIS and YT-VOS. Understandably, there are resource constraints that make it difficult to train on Kinetics to provide the apples-to-apples comparison. Still, the quantitative benchmark results in the paper could be stronger.

Overall, I lean positive on this paper. I like the idea itself, though the presentation and results are not as strong as they could be. I think that this method would likely scale up well with more resources.

Other comments:

- It's good to see the results in Table 2b, though I'm curious, given the down and up in performance from T=5->10->15 how much variance is there in the results across training runs?
- Why not include other data augmentation techniques? (L209) They are helpful signals to learn invariance to in other self-supervised work, seems like they would be of use here too.

**Final comments:**

I appreciate the effort the authors put in their rebuttal to clarify concerns around apples-to-apples comparison to prior work. I also sympathize with the headache in trying to download and use Kinetics. Overall, I'm still a fan of the proposed idea. I think the presentation and explanation of the method could be improved, but I stand by my recommendation to accept.

**Time Spent Reviewing:**

3

---

> ### Author Response · Authors · 2021-08-10
> **Response to Reviewer cHxN**
>
> Thank you for your encouraging, very detailed feedback and helpful suggestions.
>
> > The paper would benefit from a clearer presentation; rework Fig. 2.
>
> We will revise the manuscript accordingly to make the presentation clearer.
>
> > Quantitative benchmark results in the paper could be stronger.
>
> We kindly refer to the [general comment](https://openreview.net/forum?id=i8kfkuiCJCI&noteId=tLLWjX4BzSE) for a discussion of the quantitative comparison.
>
> > How much variance is there in the results across training runs?
>
> The standard deviation of the J&F score on DAVIS-2017 with a model trained on YouTube-VOS across 5 training runs is 0.42; the reported numbers are the maxima across the runs.
>
> > Why not include other data augmentation techniques?
>
> We have meanwhile experimented with two versions of data augmentation using random scaling and colour jitter:
> * Using frame-level augmentation (i.e. different augmentation per frame), we observed a decrease in VOS accuracy.
> We hypothesise that artificial augmentation techniques, such as sudden changes in contrast, saturation, or scale, poorly reflect natural transformations occurring in real-world video sequences. Using discrepant augmentation per frame may also challenge a meaningful association between the anchors, extracted from one frame, and the features from the other frames, especially at the beginning of training, on which we rely for generating the pseudo labels.
> * Using video-level augmentation (i.e. the same augmentation for every frame, but different across video clips), we did not observe a significant change in accuracy. This is expected, since the framework would be additionally required to cope with distinguishing perturbed video clips (following our implementation of Assumption 3), which does not provide useful information for VOS.

---

### Official Review · Reviewer_RkEz · 2021-07-16

**Rating:** 7
**Confidence:** 4

**Summary:**

The paper presents a method for unsupervised video object segmentation. The method is based on enforcing equivariance between video batch and the same but transformed batch. The paper utilizes  anchor sampling strategy in order to reduce computational requirements, and compute the affinity between features and anchors.

**Ethical Concerns:**

There is no ethical concerns regrading this paper.

**Limitations And Societal Impact:**

There is no specific limitations or societal impact.

**Main Review:**

Positive:
- The paper is fairly well written, easy to read and to understand.
- The paper contains thoughtful ablation studies which justify the choices made.
- The proposed method is very simple, well-thought and easy to understand.


Negative:
- The paper lacks rigor comparison with the baselines. On the Davis-2017 dataset the paper is compared with CorrFlow and MAST using the same pretraining, however the label propagation mechanism is different. On VOS it is compared with CRW, but the training datased is different. So overall there is no single clear comparison with methods from the literature on the same dataset with the same propagation strategy.  Thus it is not possible to judge clearly if the method outperforms the competitors.


Some minor staff:
- Line 17: ample -> sample

- The paper did not discuss some relevant image-based segmentation methods that also utilize equivariance, for example: SCOPS: Self-Supervised Co-Part Segmentation Hung et al.

- Why is only random scaling and horizontal flipping considered as the transformation? Did the authors experiment with other kinds of transformations: rotations, sheering, color jittering?



**Time Spent Reviewing:**

5h

---

> ### Author Response · Authors · 2021-08-10
> **Response to Reviewer RkEz**
>
> Thank you for the positive feedback and thoughtful comments.
>
>
> > Clear comparisons with methods from the literature
>
> Please, refer to our [general comment](https://openreview.net/forum?id=i8kfkuiCJCI&noteId=tLLWjX4BzSE) for a detailed discussion.
>
> > Discuss relevant image-based segmentation methods that also utilize equivariance
>
> Thank you for pointing out this relevant work. We will of course include and discuss it in the revision.
>
> > Did the authors experiment with other kinds of transformations: rotations, sheering, color jittering?
>
> We actually experimented with rotations and sheering initially, but found their benefits insignificant at the cost of increased implementation complexity.
> The main disadvantage of these transformations is the need to carefully handle the boundaries: both rotation and sheering require image padding, which needs to be removed post-hoc; otherwise, fully convolutional training would result in a degenerate solution.
> We also experimented with colour jitter and did not find it beneficial. We kindly refer to [our response to **R-cHxN**](https://openreview.net/forum?id=i8kfkuiCJCI&noteId=cfq5lGO4ckZ) for a more in-depth account of the setups we tested.

---

> > ### Comment · Reviewer_RkEz · 2021-08-29
> > **Responce to authors**
> >
> > I would like to thanks the authors for their reply, overall I find this a paper a solid submission and I want to keep my acceptance score.

---

### Official Review · Reviewer_zP7t · 2021-07-18

**Rating:** 7
**Confidence:** 4

**Summary:**

The paper proposes a self-supervised representation learning method to learn dense spatial features for videos. The goal for the representation is to be able to perform correspondences and label propagation across frames. The proposed method involves an objective that encourages features to be distinguishable on inter-frame and intra-frame levels. Through experiments on two datasets, it is shown that these features are effective for label propagation, outperforming previous methods trained with the same data and demonstrating comparable performance to methods leveraging significantly larger amounts of training data.

**Limitations And Societal Impact:**

Yes, the authors have discussed limitations and societal impact.

**Main Review:**


- The paper is extremely well written and is very easy to follow. The introduction concisely motivates the problem and presents an overview of the paper. The related work section does a thorough job of contrasting with existing work in this domain. The approach clearly presents the assumptions that are required to be satisfied for the objective to make sense.

- The presented approach involves spatially sampling features to be used as the "anchor" features. Similarity of the features from a second branch to the anchor features is used as the targets. This core idea is simple to follow and is novel (to the best of my knowledge).

- The paper does a thorough job of analyzing the implications of the presented objective (Sec 3.5) and presents motivation for all design choices (see below for some questions).

- The proposed approach is extremely efficient in training time, especially compared to related literature in self-supervised learning.

- The experimental evaluation is thorough and demonstrates superior results on two datasets compared to methods trained on the same datasets. However, the proposed representation underperforms compared to methods trained on larger datasets. Through extensive ablations, the effect of different hyperparameters are also analyzed.

Overall I think this paper makes an interesting contribution to self-supervised learning (in the context of features for label propagation), that is significantly more efficient compared to existing methods and demonstrates impressive results.

## Questions/Concerns
There are a few things that were not immediately clear and I think the paper would benefit from additional discussion on these aspects.

### Anchor Sampling
The text presents the anchor sampling as merely a method to improve efficiency. Is this really true? If all the features were indeed used, wouldn't the model become untrainable (even in the presence of enough compute)?
Because in this case the anchor affinities $v \in \mathbb{R}^{BTHW \times BTHW}$ would also have $v_{i,j} = 1$ if $i,j$ point to the same location.

### Comparison to MAST[15]
In Sec 4.1, you explain that MAST generates features at twice the resolution of your method. Does this mean that they are solving a harder correspondences/label propagation problem since they have to perform 4x the number of correspondences? If so, this doesn't entirely justify the gain in performance on DAVIS-2017 (as discussed in Line 264-266).

### DAVIS-2017 Results
In Table 1, do all the methods use the same propagation method? If not, does your method have an unfair advantage because of the new propagation method?

### YouTube-VOS Results
Similarly here, since the text claims that the MAST propagation method leverages additional information (like ROIs), is it possible to present results of MAST with your propagation method (using features before the ROI Align operator).

### Scaling up to larger datasets
CRW[13] seems to outperform your method, but uses significantly larger amount of training data. Since your proposed method is computationally efficient, is there a reason for not scaling up training to Kinetics or TrackingNet?



**Time Spent Reviewing:**

3 hours

---

> ### Author Response · Authors · 2021-08-10
> **Response to Reviewer zP7t**
>
> Thank you for the positive feedback and helpful suggestions to improve clarity.
>
> > Is the anchor sampling merely a method to improve efficiency? If all the features were indeed used, wouldn't the model become untrainable?
>
> Indeed, anchor sampling primarily serves to implement our Assumptions 1-4, and the computational efficiency comes as an additional benefit.
> If all features in a 3D volume (i.e. across space and time) become anchors, the method would potentially learn to discriminate between the same phenomena in a video (e.g. the same person in multiple frames), hence lead to inferior accuracy of dense tracking. This outcome could be inferred from our ablation setting $B=80, T=1$ in Tab. 2b, which abandons Assumption 2: the features within the same video are not pulled together, since there is only one frame per sequence in the batch.
>
>
> > Does this mean that they [MAST] are solving a harder correspondences/label propagation problem since they have to perform 4x the number of correspondences?
>
> Provided the same input resolution, the output resolution of the features produced by MAST is twice as larger as ours and that of previous work (e.g. [13]).
> This incurs additional computational (and memory) overhead both at training and inference time, and, in fact, makes the task easier, especially for small objects, since feature correlations are computed at a more fine-grained level.
> MAST also benefits from a two-stage inference process: first detecting a ROI and then computing feature correlations bounded by the ROI.
> This is contrast to the more rudimentary label propagation method used by CRW [13] and in our work, where the features between the frames are correlated without the ROI alignment.
>
>
> > DAVIS-2017 and YouTube-VOS results; scaling up to larger datasets.
>
> We kindly refer to the [general comment](https://openreview.net/forum?id=i8kfkuiCJCI&noteId=tLLWjX4BzSE) for a detailed discussion of these points.

---

### Author Response · Authors · 2021-08-10
**Response to all reviewers**

First and foremost, we sincerely thank all the reviewers for their time, helpful comments and a very positive feedback.
Here, we address the shared comment on direct comparisons from reviewers **R-zP7t**, **R-RkEz** and **R-cHxN**, and reply to the suggestions and concerns individually below.
We will incorporate all our responses in the revision.

> **[R-zP7t, R-RkEz, R-cHxN]** Clear comparisons with methods from the literature

We acknowledge that more direct comparisons would make our quantitative results stronger.
One of the complicating factors is that current state of the art, CRW [13] and MAST [15], were concurrent works at the time of their publication and were not directly comparable either, since they used both a different training dataset and a label propagation algorithm.
In our experiments, we carefully note these differences for transparency.
Nonetheless, we are keen to extend our experimental setups by:

- Using the same label propagation on DAVIS-2017 as CRW [13].
At the time of submission, we used our own implementation of the label propagation based on the details provided in [13].
We have meanwhile also tested our approach with the original label propagation of CRW to eliminate the possibility of an unfair advantage due to our own label propagation.
Using the label propagation from [13] improved our result by 0.5% and slightly exceeded that of [13]:

Training | Label propagation | J&F
--- | --- | ---
Ours | Ours | 67.3
CRW (reported) | CRW | 67.6
CRW (reproduced) | CRW | 67.5
Ours | CRW | **67.8**

For "CRW (reproduced)", we re-ran the inference with the snapshot provided in the official repository, while "CRW (reported)" corresponds to the numbers reported in [13].


- Using either the same label propagation on YouTube-VOS as MAST [15], or running MAST with our label propagation [13].
MAST is tightly coupled with the ROI-based label propagation method, since it is also employed at training time.
We did experiment with the publicly available snapshot provided by MAST and our label propagation, but were unable to produce competitive accuracy for MAST.
On the other hand, the official MAST repository provides inference only for DAVIS and [important details are missing](https://github.com/zlai0/MAST/issues/21) to exactly replicate the inference on YouTube-VOS, such as the structure of the context buffer for handling intermediate objects, or the input resolution.
We have contacted the authors again and will add this comparison, if these details are clarified.

- Scaling up to larger datasets.
We evaluate our approach using two different training datasets, YouTube-VOS and OxUvA, in line with prior work [15].
We thank for the suggestion and agree that also testing on Kinetics or TrackingNet would be useful.
In particular, we recognise that training on Kinetics-400 would enable a direct comparison to CRW [13].
Obtaining Kinetics-400, however, proved to be more time-consuming than initially expected: with its size of at least 0.5 Terabyte, the dataset videos reside in the public domain (YouTube) and are available only with limited access (bandwidth throttling and temporary suspensions).
Nevertheless, we plan to include this comparison in the final version, or over the course of the rebuttal period, if available sooner.

---

### Decision · Program_Chairs · 2021-09-27

**Decision:**

Accept (Poster)

**Comment:**

The paper discusses a method for unsupervised learning of correspondences across video frames. Different from prior work like CRW [13] the proposed method offers additional flexibility (e.g., an anchor can be matched to various points). To establish the use of this additional flexibility the reviewers asked for additional baselines which the authors provided in the rebuttal. In a discussion one reviewer remained concerned about the novelty and its use (the authors admit that the improvement compared to [13] is "slight"). Moreover, the authors also state that "testing on Kinetics or TrackingNet would be useful. In particular, we recognise that training on Kinetics-400 would enable a direct comparison to CRW [13]." Up until writing of this meta-review the authors have not provided this result. AC thinks a careful comparison to very related work is desirable and shouldn't be omitted. Further, as suggested by the reviewer that remained more concerned, AC concurs that the writing and organization of this paper should be improved significantly and AC can understand why the reviewer remained concerned. AC strongly encourages the authors to improve the paper by adding additional experimental evidence and by assessing organization.